# CRG Score: A Distribution-Aware Clinical Metric for Radiology Report Generation

**Ibrahim Ethem Hamamci**[*]                     IBRAHIM.HAMAMCI@UZH.CH

**Sezgin Er**                                    SEZGIN.ER@STD.MEDIPOL.EDU.TR

**Suprosanna Shit**                              SUPROSANNA.SHIT@UZH.CH

**Hadrien Reynaud**                              HADRIEN.REYNAUD19@IMPERIAL.AC.UK

**Bernhard Kainz**                               BERNHARD.KAINZ@FAU.DE

**Bjoern Menze**                                 BJOERN.MENZE@UZH.CH

**Editors:** Accepted for publication at MIDL 2025

## Abstract

Evaluating long-context radiology report generation is challenging. NLG metrics fail to capture clinical correctness, while LLM-based metrics often lack generalizability. Clinical accuracy metrics are more relevant but are sensitive to class imbalance, frequently favoring trivial predictions. We propose the CRG Score, a distribution-aware and adaptable metric that evaluates only clinically relevant abnormalities explicitly described in reference reports. CRG supports both binary and structured labels (e.g., type, location) and can be paired with any LLM for feature extraction. By balancing penalties based on label distribution, it enables fairer, more robust evaluation and serves as a clinically aligned reward function.

**Keywords:** report generation, evaluation metrics, 3D medical imaging, clinical accuracy

## 1. Introduction

Radiology reports are central to clinical decision-making, yet generating them—especially from 3D modalities like CT—is time-consuming due to their long and structured nature. While recent advances in volumetric modeling (Hatamizadeh et al., 2022; Hamamci et al., 2024c) and paired dataset availability (Hamamci et al., 2024a; Draelos et al., 2021) have enabled automated report generation, evaluating these reports remains a key bottleneck.

Conventional NLG metrics rely on n-gram overlap and fail to assess clinical correctness, penalizing stylistic variation while missing critical errors. Clinical accuracy (CA) metrics such as precision and recall are more appropriate but are highly sensitive to class imbalance. Accuracy often overestimates performance due to inflated true negatives, and recall may reward overgeneration. These issues are amplified in 3D reporting, where irrelevant normal findings are often omitted. LLM-based metrics offer some improvement but are typically fine-tuned for specific modalities and do not generalize well (Ostmeier et al., 2024). They also rarely account for structured or imbalanced label distributions. As open-source LLMs evolve rapidly, relying on fixed fine-tuned evaluators becomes increasingly unsustainable.

We propose the **CRG Score**, a distribution-aware and adaptable clinical accuracy metric for long-context report generation. CRG addresses key limitations by (i) avoiding report-level averaging, (ii) ignoring clinically irrelevant true negatives, (iii) balancing penalties based on abnormality distribution, and (iv) supporting both binary and structured labels. It is model-agnostic and can be paired with any recent LLM to extract report features, serving as both a robust evaluation metric and a clinically aligned reward function.

---

* Corresponding author. Implementation available at: https://github.com/ibrahimethemhamamci/CRG

## 2. Methodology

To examine the limitations of current metrics, we evaluate recent models, RadFM (Wu et al., 2023), CT2Rep (Hamamci et al., 2024b), CT-CHAT (Hamamci et al., 2024a), and Merlin (Blankemeier et al., 2024), on the CT-RATE validation set. CT2Rep and CT-CHAT were trained on CT-RATE; Merlin was re-implemented using its public code due to the lack of released weights, while RadFM was evaluated using its official pretrained model.

Table 1: Evaluation of CT report generation models using NLG metrics and Green score.

| Model | BLEU-1 | BLEU-2 | BLEU-3 | BLEU-4 | METEOR | ROUGE-L | CIDEr | Green |
|---|---|---|---|---|---|---|---|---|
| RadFM | 0.000 | 0.000 | 0.000 | 0.000 | 0.020 | 0.042 | 0.000 | 0.018 |
| CT2Rep | 0.372 | 0.292 | 0.244 | 0.213 | 0.197 | 0.362 | 0.279 | 0.487 |
| CT-CHAT | 0.373 | 0.284 | 0.231 | 0.198 | 0.215 | 0.326 | 0.199 | 0.436 |
| Merlin | 0.231 | 0.163 | 0.124 | 0.099 | 0.148 | 0.204 | 0.046 | 0.216 |

NLG metrics often overlook critical clinical content and penalize harmless lexical variation (see Table 1). Although Green (Ostmeier et al., 2024) leverages a fine-tuned LLM, it was trained on X-ray reports and generalizes poorly to CT. This underscores the need for a clinically grounded, adaptable metric that can operate with any strong feature extractor.

Table 2: Comparison of report generation using clinical accuracy metrics and CRG score.

| Model | F1 Score | Precision | Recall | Accuracy | TP | FN | FP | TN | CRG |
|---|---|---|---|---|---|---|---|---|---|
| RadFM | 0.059 | 0.170 | 0.038 | 0.786 | 550 | 9985 | 1766 | 42401 | 0.335 |
| CT2Rep | 0.160 | 0.435 | 0.128 | 0.812 | 1561 | 8974 | 1804 | 42363 | 0.359 |
| CT-CHAT | 0.184 | 0.450 | 0.158 | 0.791 | 2224 | 8311 | 3081 | 41086 | 0.368 |
| Merlin | 0.160 | 0.295 | 0.112 | 0.787 | 1504 | 9031 | 2694 | 41473 | 0.352 |

Table 2 presents CA metrics and confusion matrix elements, using CT-RATE's 18-class report labeler (Hamamci et al., 2024a). Traditional CA metrics are highly sensitive to class imbalance and often misleading: high accuracy is inflated by TNs, which may lack clinical value. RadFM shows high precision due to under-reporting (healthy bias), while CT2Rep and CT-CHAT achieve high recall at the cost of more FPs. When most labels are negative, accuracy becomes uninformative, underscoring the need for a clinically grounded metric.

### 2.1. CRG Score: A Distribution-Aware Metric

3D reports often omit normal findings unless clinically relevant. For instance, a chest CT for suspected pneumonia will typically describe the lung parenchyma but not mention a normal hiatal hernia. This aligns with the CT-RATE labeler, which infers unmentioned findings as *normal*. To address this, CRG considers only clinically meaningful outcomes.

- $T$: total number of labels in the test set = TP + FP + FN + TN
- $A$: number of positive labels (abnormalities in ground truth) in the test set
- $w_{\mathrm{TP}}$, $w_{\mathrm{FN}}$, $w_{\mathrm{FP}}$: weights for true positives, false negatives, and false positives
- $S_{\max} = A \cdot w_{\mathrm{TP}}$: max possible score, $s = \mathrm{TP} \cdot w_{\mathrm{TP}} - \mathrm{FN} \cdot w_{\mathrm{FN}} - \mathrm{FP} \cdot w_{\mathrm{FP}}$: actual score

We require the two extremes to yield equal scores: an empty report ($s = -A \cdot w_{\text{FN}}$) and an exhaustive report ($s = A \cdot w_{\text{TP}} - (T - A) \cdot w_{\text{FP}}$). Equating them gives:

$$\frac{w_{\text{TP}} + w_{\text{FN}}}{w_{\text{FP}}} = \frac{T - A}{A} \tag{1}$$

To solve the equation, we assume the reward for a TP equals the penalty for an FN ($w_{\text{TP}} = w_{\text{FN}}$), reflecting a clinically grounded trade-off. In low-prevalence settings, missing abnormalities (FNs) carries high risk and must be prioritized, even at the cost of occasional FPs. In high-prevalence cases, over-reporting (FPs) can reduce diagnostic trust. This assumption balances sensitivity and specificity according to dataset distribution:

$$\frac{w_{\text{TP}}}{w_{\text{FP}}} = \frac{T - A}{2A} \quad \Rightarrow \quad w_{\text{TP}} = w_{\text{FN}} = \frac{T - A}{2A}, \quad w_{\text{FP}} = 1 \tag{2}$$

### 2.2. Final Metric: Normalized CRG Score

Let $s$ be the model's raw score and $S_{\max}$ the maximum possible score. Then:

$$\text{CRG} = \frac{S_{\max}}{2S_{\max} - s} \tag{3}$$

This formulation yields CRG $= \frac{1}{3} = 0.\overline{3}$ for trivial solutions (always normal or always abnormal), with higher values indicating better clinical performance.

CRG is flexible across varying levels of report structure, supporting both binary abnormality labels and structured features (e.g., type, location, laterality). Researchers are not limited to predefined label sets and can extract relevant features using any recent LLM.

The example implementation in Table 2 uses 18 binary abnormalities from the CT-RATE labeler. To demonstrate CRG's adaptability, consider a two-level setup:

- CRG-1: Binary labels for general abnormality classes (e.g., *lung opacity*, *nodule*).
- CRG-2: Structured labels for lung opacity's types (such as *consolidation* or *GGO*) and location (in the left/right lung), yielding four fine-grained classes.

Each produces a CRG score, and the final score is:

$$\text{CRG}_{\text{final}} = \text{mean}(\text{CRG-1}, \text{CRG-2})$$

This method can be extended to include severity, count, or spatial attributes. CRG Score's distribution-aware design ensures fair scoring even as the granularity of the label increases.

### 2.3. Applications and Limitations

**Evaluation metric:** CRG provides a more clinically aligned evaluation for long-context report generation. Unlike existing metrics, it remains effective under class imbalance.

**Reward signal in reinforcement learning:** Most LLM-based report generation models are trained with token-level cross-entropy loss (Li et al., 2023), which poorly reflects clinical priorities. CRG can serve as a reward function to promote clinical correctness and can be combined with NLG metrics (e.g., BLEU) or cross-entropy to balance fluency and accuracy.

The main limitation of the current implementation is that CRG is evaluated only on binary abnormality labels from the CT-RATE labeler. Future work should extend it to structured attributes such as abnormality type, size, location, or count.

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
