# OpenReview forum: "CRG Score: A Distribution-Aware Clinical Metric for Radiology Report Generation"
_MIDL.io/2025/Short_Papers — MIDL 2025 - Short Papers_

### Official Review · Reviewer_pShm · 2025-04-28

**Rating:** 4
**Confidence:** 4

**Summary:**

The paper proposes a CRG score metric that is distribution-aware, adaptable, and only evaluates clinically relevant abnormalities explicitly described in reference reports.

**Strengths:**

1. It can be combined with any LLM, supports both binary and structured labels (although it only works on binary labels from CT-RATE labeler), and ignores clinically irrelevant findings.
2. Several equations are provided that describe the metric creation.

**Weaknesses:**

1. It's unclear if there is any true difference in performance between CT2Rep, CT-CHAT, and Merlin since the CRG metric is pretty much >0.35 for all.
2. A statistical analysis would be necessary to evaluate this.

---

### Decision · Program_Chairs · 2025-05-01

Accept